# TASK AFFINITY WITH MAXIMUM BIPARTITE MATCHING IN FEW-SHOT LEARNING

**Cat P. Le, Juncheng Dong, Mohammadreza Soltani, Vahid Tarokh**
Department of Electrical and Computer Engineering, Duke University

## ABSTRACT

We propose an asymmetric affinity score for representing the complexity of utilizing the knowledge of one task for learning another one. Our method is based on the maximum bipartite matching algorithm and utilizes the Fisher Information matrix. We provide theoretical analyses demonstrating that the proposed score is mathematically well-defined, and subsequently use the affinity score to propose a novel algorithm for the few-shot learning problem. In particular, using this score, we find relevant training data labels to the test data and leverage the discovered relevant data for episodically fine-tuning a few-shot model. Results on various few-shot benchmark datasets demonstrate the efficacy of the proposed approach by improving the classification accuracy over the state-of-the-art methods even when using smaller models.

## 1 INTRODUCTION

Leveraging the knowledge of one task in training the other related tasks is an effective approach to training deep neural networks with limited data. In fact, transfer learning, multi-task learning (Standley et al., 2020), and meta-learning (Finn et al., 2017) are examples of training a new task using the knowledge of others. In fact, a strong piece of work (Standley et al., 2020) has shown that training similar tasks together in multi-task learning often achieves higher accuracy on average. However, characterizing the similarity between tasks remains a challenging problem. In this paper, we present a task similarity measure representing the complexity of utilizing the knowledge of one task for learning another one. Our measure, called *Task Affinity Score* (TAS), is non-commutative and is defined as a function of the Fisher Information matrix, which is based on the second-derivative of the loss function with respect to the parameters of the model under consideration. By definition, the TAS between two tasks is always greater or equal to 0, where the equality holds if and only if both tasks are identical. For the classification tasks, the TAS is invariant to the permutation of the data labels. In other words, modifying the numeric order of the data labels does not affect the affinity score between tasks. Additionally, TAS is mathematically well-defined, as we will prove in the sequel.

Following the introduction of TAS, we propose a few-shot learning method based on the similarity between tasks. The lack of sufficient data in the few-shot learning problem has motivated us to use the knowledge of similar tasks for our few-shot learning method. In particular, our approach is capable of finding the relevant training labels to the ones in the given few-shot target tasks, and utilizing the corresponding data samples for episodically fine-tuning the few-shot model. Similar to recent few-shot approaches (Chen et al., 2021; Tian et al., 2020), we first use the entire training dataset to train a Whole-Classification network. Next, this trained model is used for extraction of the feature vectors for a set of constructed source task(s) generated from the training dataset. The purpose of the sources task(s) is to establish the most related task(s) to the target task defined according to the test data. In our framework, TAS with a graph matching algorithm is applied to find the affinity scores and the identification of the most related source task(s) to the target task. Lastly, we follow the standard few-shot meta-learning in which a set of base tasks are first constructed, and a few-shot model is fine-tuned according to the query set of these base tasks. Our approach has a unique distinguishing property from the common meta-learning approaches: our base tasks are constructed only based on the previously discovered related source tasks to episodically fine-tune the few-shot model. Specifically, the feature vectors of the query data from the base tasks are extracted by the encoder of the Whole-Classification network, and a k-nearest neighbors (k-NN) is applied to classify the features into the correct classes by updating the weights in the encoder.

Using extensive simulations, we demonstrate that our approach of utilizing only the related training data is an effective method for boosting the performance of the few-shot model with less number of parameters in both 5-way 1-shot and 5-way 5-shot settings for various benchmark datasets. Experimental results on miniImageNet (Vinyals et al., 2016), tieredImageNet (Ren et al., 2018), CIFAR-FS (Bertinetto et al., 2018), and FC-100 (Oreshkin et al., 2018) datasets are provided demonstrating the efficacy of the proposed approach compared to other state-of-the-art few-shot learning methods.

## 2 RELATED WORK

The similarity between tasks has been mainly studied in the transfer learning literature. Many approaches in transfer learning (Silver & Bennett, 2008; Finn et al., 2016; Mihalkova et al., 2007; Niculescu-Mizil & Caruana, 2007; Luo et al., 2017; Razavian et al., 2014; Pan & Yang, 2010; Mallya & Lazebnik, 2018; Fernando et al., 2017; Rusu et al., 2016; Zamir et al., 2018; Kirkpatrick et al., 2017; Chen et al., 2018) are based on the assumption that similar tasks often share similar architectures. However, these works mainly focus on transferring the trained weights from the previous tasks to an incoming task, and do not seek to define the measurement that can identify the related tasks. Though the relationship between visual tasks has been recently investigated by various papers (Zamir et al., 2018; Pal & Balasubramanian, 2019; Dwivedi & Roig, 2019; Achille et al., 2019; Wang et al., 2019; Standley et al., 2020), these works only focus on the weight-transferring and do not use task similarity for discovering the closest tasks for improving the overall performance. Additionally, the measures of task similarity from these papers are often assumed to be symmetric, which is not typically a realistic assumption. For example, it is easier to utilize the knowledge of a comprehensive task for learning a simpler task than the other way around.

In the context of the few-shot learning (FSL), the task affinity (similarity) has not been explicitly considered. Most of the recent few-shot learning approaches are based on the meta-learning frameworks (Santoro et al., 2016; Finn et al., 2017; Vinyals et al., 2016; Snell et al., 2017). In these approaches, episodic learning is often used in the training phase, in which the FSL models are exposed to data episodes. Each episode, consisting of the support and query sets, is characterized by the number of classes, the number of samples per class in the support set, and the number of samples per class in the query set. During the training phase, the loss over these training episodes is minimized. Generally, these episodic learning approaches can be divided into three main categories: metric-based method, optimization-based method, and memory-based method. In metric-based methods (Vinyals et al., 2016; Snell et al., 2017; Koch et al., 2015; Sung et al., 2018), a kernel function learns to measure the distance between data samples in the support sets, then classifies the data in the query set according to the closest data samples in the support set. On the other hand, the goal of optimization-based methods (Finn et al., 2017; Grant et al., 2018; Rusu et al., 2018; Lee et al., 2019; Nichol et al., 2018) is to find the models with faster adaption and convergence. Lastly, the memory-based methods (Santoro et al., 2016; Ravi & Larochelle, 2017; Munkhdalai et al., 2018) use the network architectures with memory as the meta-learner for the few-shot learning. Overall, episodic learning in FSL has achieved great success on various few-shot meta-datasets.

Recently, several methods with a pre-trained Whole-Classification network have achieved state-of-the-art performance on multiple FSL benchmarks (Chen et al., 2021; Tian et al., 2020; Rodríguez et al., 2020; Rizve et al., 2021). Instead of initializing the FSL model from scratches, these methods focus on leveraging the entire training labels for pre-training a powerful and robust classifier. The pre-trained model, by itself, outperforms several meta-learning approaches in numerous FSL datasets (e.g., miniImageNet, tieredImageNet, CIFAR-FS, etc.). Next, the Whole-Classification network is used as a feature extractor for a simple base learner (e.g., logistic regression, K-nearest neighbor, etc.) and is often fine-tuned using episodic learning. Various efforts have been investigated to improve the Whole-Classification network for the few-shot learning, including manifold mixup as self-supervised loss (Mangla et al., 2020), knowledge distillation on the pre-trained classifier (Verma et al., 2019; Tian et al., 2020), and data-augmentation with combined loss functions (Rodríguez et al., 2020; Rizve et al., 2021). However, none of these approaches consider the task affinity in their training procedure. Here, we propose a task affinity measure that can identify the related tasks to a target task, given only a few data samples. Then we utilize the data samples in the related tasks for episodically fine-tuning the final few-shot classifier.

## 3 PRELIMINARIES

In this section, we present the definition of the task affinity score. First, we need to define some notations and definitions used throughout this paper. We denote the matrix infinity-norm by $||B||_\infty = \max_{i,j} |B_{ij}|$. We also denote a task $T$ and its dataset $X$ jointly by a pair $(T, X)$. To be consistent with the few-shot terminologies, a dataset $X$ is shown by the union of the *support* set, $X^{support}$ and the *query* set, $X^{query}$, i.e., $X = X^{support} \cup X^{query}$. Let $\mathcal{P}_{N_\theta}(T, X^{query}) \in [0, 1]$ be a function that measures the performance of a given model $N_\theta$, parameterized by $\theta \in \mathbb{R}^d$ on the query set $X^{query}$ of the task $T$. We define an $\varepsilon$-approximation network, representing the task-dataset pair $(T, X)$ as follows:

**Definition 1** ($\varepsilon$-approximation Network). *A model $N_\theta$ is called an $\varepsilon$-approximation network for a pair task-dataset $(T, X)$ if it is trained using the support data $X^{support}$ such that $\mathcal{P}_{N_\theta}(T, X^{query}) \geq 1 - \varepsilon$, for a given $0 < \varepsilon < 1$.*

In practice, the architectures for the $\varepsilon$-approximation networks for a given task $T$ are selected from a pool of well-known hand-designed architectures, such as ResNet, VGG, DenseNet, etc. We also need to recall the definition of the Fisher Information matrix for a neural network.

**Definition 2** (Fisher Information Matrix). *For a neural network $N_\theta$ with weights $\theta$, data $X$, and the negative log-likelihood loss function $L(\theta) := L(\theta, X)$, the Fisher Information matrix is defined as:*

$$F(\theta) = \mathbb{E}\Big[\nabla_\theta L(\theta)\nabla_\theta L(\theta)^T\Big] = -\mathbb{E}\Big[\mathbf{H}\big(L(\theta)\big)\Big], \tag{1}$$

*where $\mathbf{H}$ is the Hessian matrix, i.e., $\mathbf{H}\big(L(\theta)\big) = \nabla_\theta^2 L(\theta)$, and expectation is taken w.r.t the data.*

In practice, we use the empirical Fisher Information matrix computed as follows:

$$\hat{F}(\theta) = \frac{1}{|X|} \sum_{i \in X} \nabla_\theta L^i(\theta)\nabla_\theta L^i(\theta)^T, \tag{2}$$

where $L^i(\theta)$ is the loss value for the $i^{\text{th}}$ data point in $X$. Next, we define the task affinity score, which measures the similarity from a source task, $T_a$ to a target task, $T_b$.

**Definition 3** (Task Affinity Score (TAS)). *Let $(T_a, X_a)$ be the source task-dataset pair with $N_{\theta_a}$ denotes its corresponding $\varepsilon$-approximation network. Let $F_{a,a}$ be the Fisher Information matrix of $N_{\theta_a}$ with the query data $X_a^{query}$ from the task $T_a$. For the target task-dataset pair $(T_b, X_b)$, let $F_{a,b}$ be the Fisher Information matrix of $N_{\theta_a}$ with the support data $X_b^{support}$ from the task $T_b$. We define the TAS from the source task $T_a$ to the target task $T_b$ based on Fréchet distance as follows:*

$$s[a, b] := \frac{1}{\sqrt{2}} Trace\Big( F_{a,a} + F_{a,b} - 2(F_{a,a}F_{a,b})^{1/2} \Big)^{1/2}. \tag{3}$$

Here, we use the diagonal approximation of the Fisher Information matrix since computing the full Fisher matrix is prohibitive in the huge space of neural network parameters. We also normalize these matrices to have unit trace. As a result, the TAS in equation (3) can be simplified by the following formula:

$$s[a, b] = \frac{1}{\sqrt{2}} \left\| F_{a,a}^{1/2} - F_{a,b}^{1/2} \right\|_F = \frac{1}{\sqrt{2}} \left[ \sum_i \left( (F_{a,a}^{ii})^{1/2} - (F_{a,b}^{ii})^{1/2} \right)^2 \right]^{1/2}, \tag{4}$$

where $F^{ii}$ denotes the $i^{\text{th}}$ diagonal entry of the Fisher Information matrix. The TAS ranges from 0 to 1, with the score $s = 0$ denotes a perfect similarity and the score $s = 1$ indicates a perfect dissimilarity. In the next section, we present our few-shot approach based on the above TAS.

## 4 FEW-SHOT APPROACH

In the few-shot learning problem, a few-shot task of $M$-way $K$-shot is defined as the classification of $M$ classes, where each class has only $K$ data points for learning. One common approach (sometimes called *meta-learning*) to train a model for $M$-way $K$-shot classification is to first construct some training tasks from the training dataset, such that each task has a support set with $M \times K$

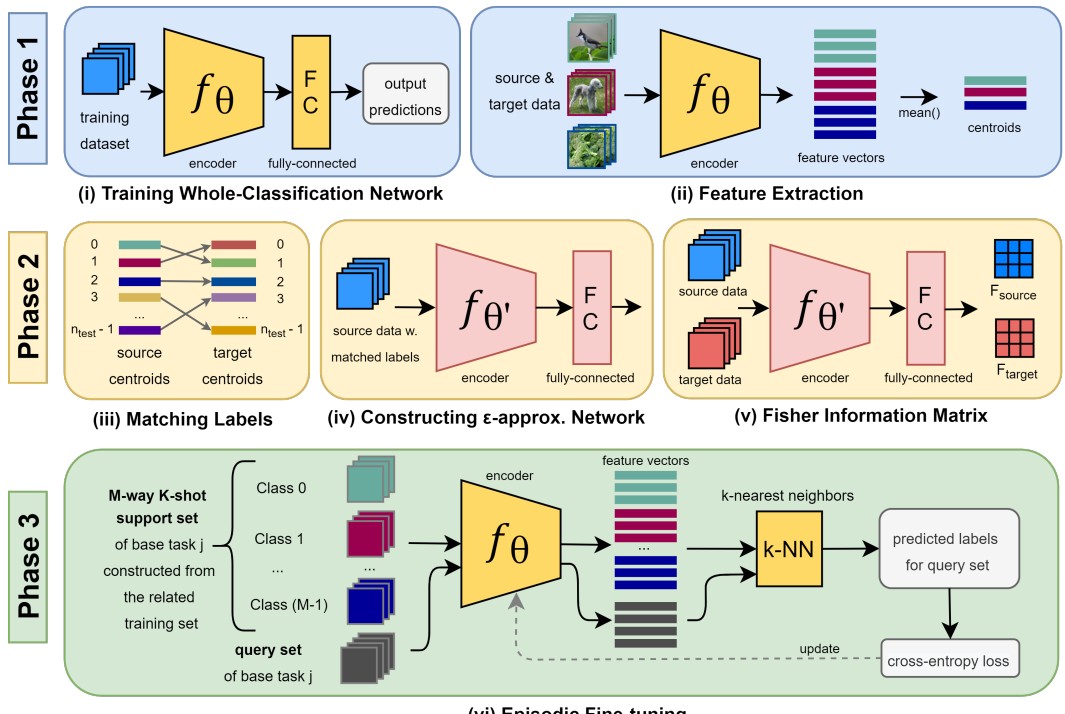

Figure 1: The overview of our proposed few-shot learning approach. **(i)** Train the whole-classification network with the entire labels from the training dataset. **(ii)** Use the encoder $f_\theta$ to extract the feature vectors for each class in the source tasks and the target task, and compute their mean (or centroid). **(iii)** Maximum matching algorithm is applied to map the source task's centroids to the target task's centroids. **(iv)** Construct the $\varepsilon$-approximation network for the source task(s), and using the modified dataset of the source task(s) to train the $\varepsilon$-approximation network. **(v)** Obtain the Fisher Information matrices for the source task(s) and the target task, and computing the TAS between them. The source tasks with the smallest TAS are considered as related tasks. **(vi)** From the related-training set (consists of the data samples from the related tasks), generate few-shot $M$-way $K$-shot base tasks. Use the support and query sets of the base tasks to episodically fine-tune $f_\theta$.

samples and a separate query set of data points with the same $M$ labels in the support set. The goal is to use the training data (the support set for training and the query set for evaluating the performance of the training tasks), and the support set of the test data to train a model to achieve high performance on the query set of the test data. Recently, there is another few-shot approach called *Whole-Classification* (Chen et al., 2021; Tian et al., 2020) is proposed to use the whole training dataset for training a base classifier with high performance. Here the assumption is that the training set is a large dataset, which is sufficient for training a high accuracy classifier with all the labels in the training set. On the other hand, the test set, which is used to generate the few-shot test tasks, is not sufficiently large to train a classifier for the entire labels. Here, our proposed few-shot learning approach, whose pseudo-code is shown in Algorithm 1, consists of three phases:

1. **Training Whole-Classification Network and Feature Extraction.** In phase 1 (steps (i) and (ii) in Figure 1), we construct a Whole-Classification network to represent the entire training set. To do so, we use the standard ResNet-12 as a common backbone used in many few-shot approaches for constructing the Whole-Classification network. Next, we train this network to classify the whole classes in the training set. Additionally, the knowledge distillation technique can be applied in order to improve the classification accuracy and generalization of the extracted features (Sun et al., 2019). Next, we define a set of few-shot training tasks (i.e.source tasks) and a target task with the number of classes equals to the number of labels in the test data. Using the encoder of the trained Whole-Classification network, we extract the embedded features and compute the mean of each label.

---

**Algorithm 1:** Few-Shot Learning with Task Affinity Score

---

**Data:** $(X_{train}, y_{train}, n_{train}), (\{X_{test}^{support} \cup X_{test}^{query}\}, y_{test}, n_{test})$
**Input:** The Whole-Classification network $N_\theta$
**Output:** Few-shot classifier model $f_{\theta^*}$

1 **Function** mTAS $(X_a, C_a, X_b, C_b, N_{\theta_a})$:
2      Obtain the mapping order: $mapping = \texttt{MaximumMatching}(C_a, C_b)$
3      Fine-tune $N_{\theta_a}$ using $X_a$ with mapped labels (building the $\varepsilon$-approximation network)
4      Compute $F_{a,a}, F_{a,b}$ using $N_{\theta_a}$ with $X_a, X_b$, respectively
5      **return** $s[a, b] = \dfrac{1}{\sqrt{2}} \left\| F_{a,a}^{1/2} - F_{a,b}^{1/2} \right\|_F$

6 **Function** Main:
7      Train $N_\theta$ with the entire $n_{train}$ labels in $X_{train}$      ▷ Beginning of phase 1
8      Construct $S$ source tasks, where each has $n_{test}$ labels from $X_{train}$
9      Construct target task using $n_{test}$ labels in $X_{test}^{support}$
10      Extract set $C_{source}$ of class centroids for each source task using the encoder $f_\theta$ of $N_\theta$
11      Extract set $C_{target}$ of class centroids for the target task using the encoder $f_\theta$ of $N_\theta$
                                                           ▷ Beginning of phase 2
12      **for** $i = 1, 2, \ldots, S$ **do**
13          $s_i = \texttt{mTAS}(X_{source_i}, C_{source_i}, X_{test}^{support}, C_{target}, N_\theta)$
14      **return** closest tasks: $i^* = \underset{i}{\operatorname{argmin}}\ s_i$
15      Define set $L$ which consists of labels from closest tasks      ▷ Beginning of phase 3
16      Define a subset data $X_{related} = \{X_{train} | y_{train} \in L\}$
17      Construct $M$-way $K$-shot base tasks and fine-tune $f_\theta$ episodically using $X_{related}$

---

    2. **Task Affinity.** Using the computed mean of the feature vectors from the previous phase, phase 2 (steps (iii), (iv) and (v) in Figure 1) identifies the closest source tasks to the target task by applying the TAS and a pre-process step called matching labels.

    3. **Episodic Fine-tuning.** After obtaining the closest source tasks, we construct a training dataset called *related-training* set, consisting of samples only from the labels in the closest source tasks. In phase 3 (step (vi) in Figure 1), we follow the standard meta-learning approach by constructing the $M$-way $K$-shot base tasks **from the related-training set** to episodically fine-tune the classifier model. Since the classifier model is fine-tuned only with the labels of the related-training set, the complexity of training of the few-shot model is reduced; hence, the overall performance of the few-shot classification increases.

## 4.1 PHASE 1 - TRAINING WHOLE-CLASSIFICATION NETWORK AND FEATURE EXTRACTION

Recent few-shot learning approaches (e.g., Chen et al. (2021); Tian et al. (2020)) have shown advantages of a pre-trained Whole-Classification network on the entire training dataset. In this paper, we train a classifier network, $N_\theta$ using the standard architecture of ResNet-12 with the entire training classes. The Whole-Classification network $N_\theta$ can be described as concatenation of the encoder $f_\theta$ and the last fully-connected layer (illustrated in Figure 1(i)). Since $N_\theta$ is trained on the entire training set $X_{train}$, it can extract meaningful hidden features for the data samples in the training set. Next, we construct a target task using $n_{test}$ classes and its corresponding data points in the support set of the test data (i.e., $X_{test}^{support}$). We also construct a set of $S$ few-shot source tasks using the training set, $X_{train}$, where each task has $n_{test}$ labels sampled from $n_{train}$ classes of the training set. This results in a set of source tasks with their corresponding training data, i.e., $(T_{source_i}, X_{source_i})$ for $i = 1, 2 \ldots, S$. Using the encoder part of $N_\theta$ denoted by $f_\theta$, we extract the feature centroids from $X_{source_i}$ and $X_{test}^{support}$. Finally, we compute the mean (or centroid) of the extracted features for every class in $X_{source_i}$ of the source task $T_{source_i}$, and similarly for $X_{test}^{support}$ of the target task. That is, we compute $\frac{1}{|X_c|} \sum_{x \in X_c} f_\theta(x)$, where $X_c$ denotes the set of data samples belonging to class $c$ in either source tasks, or the target task. In phase 2, we use the computed centroids to find the best way to match the classes in source tasks to the target task.

### 4.2 PHASE 2 - TASK AFFINITY

In this phase, our goal is to find the most related source task(s) in the training set to the few-shot target task in the test set. To do this end, we apply the proposed task affinity score. However, naively applying the TAS on the extracted centroids may result in finding a non-related source task(s) to the target task. This is because neural networks are not invariant to the label's permutation (Zhang et al., 2021). For instance, consider task I of classifying cat images as 0 and dog images as 1 and consider the $\varepsilon$-approximation network $N_{\theta_I}$ with zero-error (i.e., $\varepsilon = 0$). Also, let task J be defined as the classification of the dog images encoded as 0 and cat images as 1. Assume that the dataset used for both tasks I and J is the same. If we use $N_{\theta_I}$ to predict the data from task J, the prediction accuracy will be zero. For human perception, these tasks are the same. However, the neural network $N_{\theta_I}$ can only be a good approximation network for task I, and not for task J. A naive approach to overcome this issue is to try all permutations of labels. However, it is not a practical approach, due to the intense computation requirement. As a result, we propose a method of pre-processing the labels in the source tasks to make the task affinity score invariant to the permutation of labels.

#### 4.2.1 MAXIMUM BIPARTITE MATCHING

We use the Maximum Bipartite Graph Matching algorithm to match each set of $n_{test}$ centroids of source tasks to the ones in the target task based on the Euclidean distance from each centroid to another. The goal is to find a matching order so that the total distance between all pairs of centroids to be minimized. We apply the Hungarian maximum matching algorithm (Kuhn, 1955), as shown in Figure 1(iii). The result of this algorithm is the way to match labels in the source tasks to the classes in the target task. After the matching, we modify the labels of the source tasks according to the matching result and use them to construct an $\varepsilon$-approximation network $N_{\theta'}$ for the source task, as marked (iv) in Figure 1. Please note that the weights are fine-tuned using the modified matched labels; hence, we use $\theta'$ instead of $\theta$ for the parameters of the $\varepsilon$-approximation network.

#### 4.2.2 TASK AFFINITY SCORE

Now, we can find the closest source tasks to the target task. To do so, for each source task, we compute two Fisher Information matrices (Figure 1(v)) by passing the source tasks' and target task's datasets through the $\varepsilon$-approximation network $N_{\theta'}$, and compute the task affinity score defined in the equation (4) between two tasks. Since we use the matched labels of source tasks w.r.t. the target task, the resulted TAS is consistent and invariant to the permutation of labels. The helper function `mTAS()` in the Algorithm 1 implements all the steps in the second phase. Finally, the top-$R$ scores and their corresponding classes are selected. These classes and their corresponding data points are considered as the closest source task(s) to the target task. Next, we show that our definition of the task affinity score is mathematically well-defined under some assumptions. First, from the equation (4), it is clear that the TAS of a task from itself always equals zero. Now assume that the objective function corresponding to the $\varepsilon$-approximation network is strongly convex. In the Theorem 1, we show that the TAS from task A to task B calculated using the Fisher Information matrix of the approximation network of task A on different epochs of the SGD algorithm converges to a constant value given by the TAS between the above tasks when the Fisher Information matrix is computed on the global optimum of the approximation network[1].

**Theorem 1.** *Let $X_A$ be the dataset for the task $T_A$ with the objective function $L$, and $X_B$ be the dataset for the task $T_B$. Assume $X_A$ and $X_B$ have the same distribution. Consider an $\varepsilon$-approximation network $N_\theta$ for the pair task-dataset $(T_A, X_A)$ with the objective functions $L$ and with the SGD algorithm to result weights $\theta_t$ at time $t$. Assume that the function $L$ is strongly convex, and its 3rd-order continuous derivative exists and bounded. Let the noisy gradient function in training $N_\theta$ network using SGD algorithm be given by: $g(\theta_t, \epsilon_t) = \nabla L(\theta_t) + \epsilon_t$, where $\theta_t$ is the estimation of the weights for network $N$ at time $t$, and $\nabla L(\theta_t)$ is the true gradient at $\theta_t$. Assume that $\epsilon_t$ satisfies $\mathbb{E}[\epsilon_t | \epsilon_0, ..., \epsilon_{t-1}] = 0$, and satisfies $s = \lim_{t \to \infty} \left|\left|[\epsilon_t \epsilon_t^T | \epsilon_0, \ldots, \epsilon_{t-1}]\right|\right|_\infty < \infty$ almost surely (a.s.). Then the task affinity score between $T_A$ and $T_B$ computed on the average of estimated*

---

[1]As mentioned in the Theorem 1, we assume that loss function of the approximation network is strongly convex; hence, it has a global optimum.

*weights up to the current time $t$ converges to a constant as $t \to \infty$. That is,*

$$s_t = \frac{1}{\sqrt{2}} \left\| \bar{F}_{A_t}^{1/2} - \bar{F}_{B_t}^{1/2} \right\|_F \xrightarrow{\mathcal{D}} \frac{1}{\sqrt{2}} \left\| F_A^{*1/2} - F_B^{*1/2} \right\|_F, \tag{5}$$

*where $\bar{F}_{A_t} = F(\bar{\theta}_t, X_A^{query})$, $\bar{F}_{B_t} = F(\bar{\theta}_t, X_B^{support})$ with $\bar{\theta}_t = \frac{1}{t} \sum_t \theta_t$. Moreover, $F_A^* = F(\theta^*, X_A^{query})$ and $F_B^* = F(\theta^*, X_B^{support})$ denote the Fisher Information matrices computed using the optimum approximation network (i.e., a network with the global minimum weights, $\theta^*$).*

To prove Theorem 1 (please see the appendix for the complete proof), we use the Polyak & Juditsky (1992) theorem for the convergence of the SGD algorithm for strongly convex functions. Although the loss function in a deep neural network is not strongly convex, establishing the fact that the TAS is mathematically well-defined for this case is an important step towards the more general deep neural networks and a justification for the success of our empirical observations. Furthermore, some recent works try to generalize Polyak & Juditsky (1992) theorem for the convex or even some non-convex functions in an (non)-asymptotic way (Gadat & Panloup, 2017). However, we do not pursue these versions here.

### 4.3 Phase 3 - Episodic Fine-tuning

After obtaining the closest source tasks, we construct a training dataset called *related-training* set, consisting of samples only from the labels in the closest source tasks. Following the standard meta-learning approach, we consider various $M$-way $K$-shot base tasks from the related-training dataset, we fine-tune the encoder $f_\theta$ from the original Whole-Classification network in phase 1. To be more precise, we extract the feature vectors from data samples in the support set and the query set of the base task(s). The embedded features from the support set serve as the training data for the k-NN classifier. We utilize the k-NN model to predict the labels for the data samples in the query set of the base task under consideration using their feature vectors. Finally, the cross-entropy loss is used to update the encoder $f_\theta$ by comparing the predicting labels and the labels of the query set, and backpropagating the error to the model $f_\theta$. Then this process can be repeated for several base tasks until $f_\theta$ achieves a high accuracy[2]. Figure 1(vi) shows the above process for the $j^{th}$ base task.

## 5 Experimental Study

In this section, we present our experimental results in various few-shot learning benchmarks, including miniImageNet (Vinyals et al., 2016), tieredImageNet (Ren et al., 2018), CIFAR-FS (Bertinetto et al., 2018), and FC-100 (Oreshkin et al., 2018) [3]. The miniImageNet dataset consists of $100$ classes, sampled from ImageNet (Russakovsky et al., 2015), and randomly split into $64, 16$, and $20$ classes for training, validation, and testing, respectively. Similarly, the tieredImageNet dataset is also a derivative of ImageNet, containing a total of $608$ classes from $34$ categories. It is split into $20, 6$, and $8$ categories (or $351, 97$, and $160$ classes) for training, validation, and testing, respectively. Each class, in both miniImageNet and tieredImageNet, includes $600$ colorful images of size $84 \times 84$.

We use the architecture of ResNet-12 as the Whole-Classification network, which consists of $4$ residual blocks, each block has $3$ convolutional layers and a max-pooling layer. There are different variations of ResNet-12 in the literature, each with different size of parameters (e.g., $[64, 128, 256, 512]$, $[64, 160, 320, 640]$). In our experiments, we report the number of parameters for all the models we use and for those ones compare with. For training the Whole-Classification network, the SGD optimizer is applied with the momentum of $0.9$, and the learning rate is initialized at $0.05$ with the decay factor of $0.1$ for all experiments. In miniImageNet experiment, we train the model for $100$ epochs with a batch size of $64$, and the learning rate decaying at epoch $90$. For tieredImageNet, we train for $120$ epochs with a batch size of $64$, and the learning rate decaying two times at epochs $40$ and $80$.

For miniImageNet, we define $S = 2000$ source tasks by randomly drawing samples from $20$ labels (number of labels of the target task) out of $64$ training labels from the whole training set. Next, we use the Whole-Classification network to extract the feature vectors from $S$ source tasks, and match them with labels of the target task. Next, we construct the $\varepsilon$-approximation network using the

---

[2]The final fine-tuned few-shot model is denoted by $f_{\theta*}$ in the Algorithm 1.

[3]Due to page limits, we discuss the experiments on CIFAR-FS and FC-100 in the appendix.

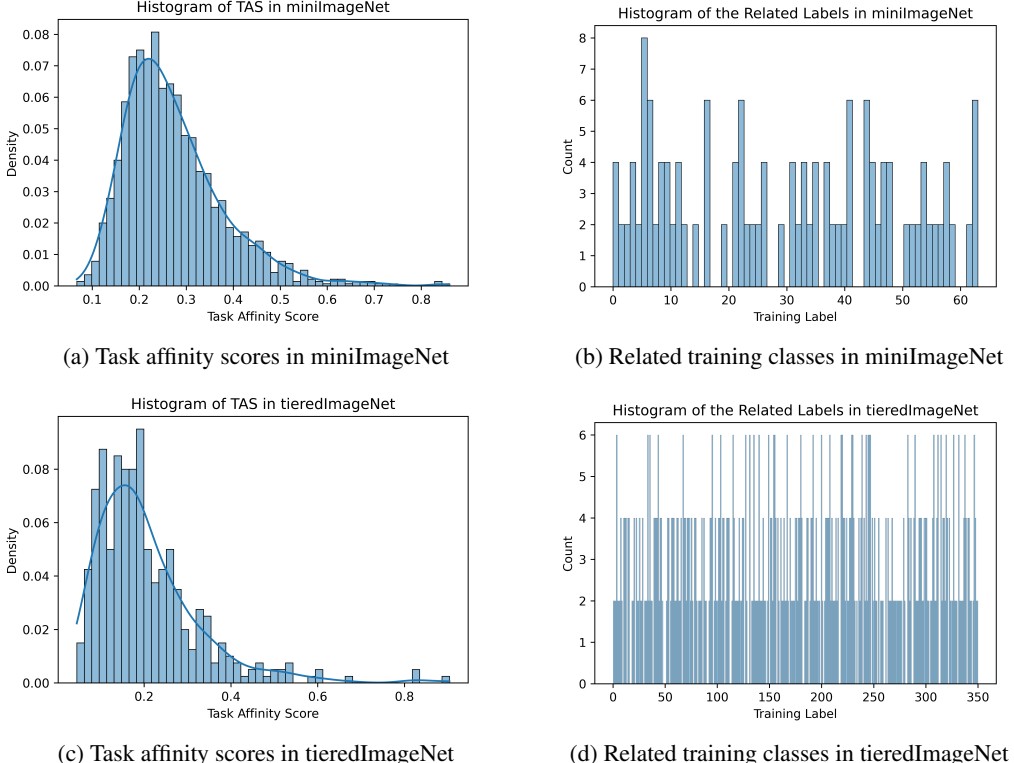

(a) Task affinity scores in miniImageNet      (b) Related training classes in miniImageNet

(c) Task affinity scores in tieredImageNet      (d) Related training classes in tieredImageNet

Figure 2: (a) The distribution of TAS found in miniImageNet. (b) The frequency of $64$ classes in the top-$8$ closest source tasks in miniImageNet. (c) The distribution of TAS found in tieredImageNet. (d) The frequency of $351$ classes in the top-$6$ closest source tasks in tieredImageNet.

encoder $f_\theta$ of the Whole-Classification network, and use it to compute the TAS in the task affinity phase. The distribution of the TAS found from the $2000$ generated source tasks in the miniImageNet is shown in Figure 2a. As we can seen, the probability distribution of the task affinity is not a non-informative or uniform distribution. Only a few source tasks, with particular sets of labels, can achieve the small TAS and are considered as the related tasks. Now, we plot the frequency of all $64$ labels in the training set which exist in the top-$8$ closest source tasks in Figure 2b. This shows that the class with the label $6$ has $8$ samples in the top-$8$ closest sources task, and this class is the most relevant one to the labels in the target task. Moreover, there are only $49$ unique classes that construct the top-$8$ closest source tasks. This means that there are some classes which do not have any samples among the top-$8$ closest sources tasks. These labels are the ones with the least similarity to the labels in target task. Similarly, we define $S = 1000$ source tasks in tieredImageNet by randomly drawing samples from $160$ labels out of entire $351$ training labels. The distribution of the TAS from these $1000$ generated source tasks is shown in Figure 2c. Similar to the miniImageNet case, Figure 2d illustrates the frequency of the $351$ training classes in the top-$6$ closest source tasks. Among $351$ training classes, there are only $293$ unique classes that construct the top-$6$ closest source tasks.

In the episodic fine-tuning phase, we use the SGD optimizer with momentum $0.9$, and the learning rate is set to $0.001$. The batch size is set to $4$, where each batch of data includes $4$ few-shot tasks. The loss is defined as the average loss among few-shot tasks. After we trained our few-shot model, we tested it on the query sets of test tasks. Table 1 and Table 2 show the performance of our approach in 5-way 1-shot and 5-way 5-shot on miniImageNet and tieredImageNet datasets, respectively. In both cases, our approach with the standard ResNet-12 is comparable to or better than IE-distill (Rizve et al., 2021), while outperforming RFS-distill (Tian et al., 2020), EPNet (Rodríguez et al., 2020) and other methods with a significantly smaller classifier model in terms of the number of parameters. Note that, EPNet (Rodríguez et al., 2020) and IE-distill (Rizve et al., 2021) also utilize the data augmentation during pre-training for better feature extraction.

Table 1: Comparison of the accuracy against state-of-the art methods for 5-way 1-shot and 5-way 5-shot classification with 95% confidence interval on miniImageNet dataset.

| Model | Backbone | Params | 1-shot | 5-shot |
|---|---|---|---|---|
| Matching-Net (Vinyals et al., 2016) | ConvNet-4 | 0.11M | $43.56_{\pm0.84}$ | $55.31_{\pm0.73}$ |
| MAML (Finn et al., 2017) | ConvNet-4 | 0.11M | $48.70_{\pm1.84}$ | $63.11_{\pm0.92}$ |
| Prototypical-Net (Snell et al., 2017) | ConvNet-4 | 0.11M | $49.42_{\pm0.78}$ | $68.20_{\pm0.66}$ |
| Simple CNAPS (Bateni et al., 2020) | ResNet-18 | 11M | $53.2_{\pm0.90}$ | $70.8_{\pm0.70}$ |
| Activation-Params (Qiao et al., 2018) | WRN-28-10 | 37.58M | $59.60_{\pm0.41}$ | $73.74_{\pm0.19}$ |
| LEO (Rusu et al., 2018) | WRN-28-10 | 37.58M | $61.76_{\pm0.08}$ | $77.59_{\pm0.12}$ |
| Baseline++ (Chen et al., 2019) | ResNet-18 | 11.17M | $51.87_{\pm0.77}$ | $75.68_{\pm0.63}$ |
| SNAIL (Mishra et al., 2017) | ResNet-12 | 7.99M | $55.71_{\pm0.99}$ | $68.88_{\pm0.92}$ |
| AdaResNet (Munkhdalai et al., 2018) | ResNet-12 | 7.99M | $56.88_{\pm0.62}$ | $71.94_{\pm0.57}$ |
| TADAM (Oreshkin et al., 2018) | ResNet-12 | 7.99M | $58.50_{\pm0.30}$ | $76.70_{\pm0.30}$ |
| MTL (Sun et al., 2019) | ResNet-12 | 8.29M | $61.20_{\pm1.80}$ | $75.50_{\pm0.80}$ |
| MetaOptNet (Lee et al., 2019) | ResNet-12 | 12.42M | $62.64_{\pm0.61}$ | $78.63_{\pm0.46}$ |
| SLA-AG (Lee et al., 2020) | ResNet-12 | 7.99M | $62.93_{\pm0.63}$ | $79.63_{\pm0.47}$ |
| ConstellationNet (Xu et al., 2020) | ResNet-12 | 7.99M | $64.89_{\pm0.23}$ | $79.95_{\pm0.17}$ |
| RFS-distill (Tian et al., 2020) | ResNet-12 | 13.55M | $64.82_{\pm0.60}$ | $82.14_{\pm0.43}$ |
| EPNet (Rodríguez et al., 2020) | ResNet-12 | 7.99M | $65.66_{\pm0.85}$ | $81.28_{\pm0.62}$ |
| Meta-Baseline (Chen et al., 2021) | ResNet-12 | 7.99M | $63.17_{\pm0.23}$ | $79.26_{\pm0.17}$ |
| IE-distill[1] (Rizve et al., 2021) | ResNet-12 | 9.13M | $65.32_{\pm0.81}$ | $83.69_{\pm0.52}$ |
| **TAS-simple (ours)** | **ResNet-12** | **7.99M** | **$64.71_{\pm0.43}$** | **$82.08_{\pm0.45}$** |
| **TAS-distill (ours)** | **ResNet-12** | **7.99M** | **$65.13_{\pm0.39}$** | **$82.47_{\pm0.52}$** |
| **TAS-distill[2] (ours)** | **ResNet-12** | **12.47M** | **$65.68_{\pm0.45}$** | **$83.92_{\pm0.55}$** |

[1] performs with standard ResNet-12 with Dropblock as a regularizer, [2] performs with wide-layer ResNet-12

Table 2: Comparison of the accuracy against state-of-the art methods for 5-way 1-shot and 5-way 5-shot classification with 95% confidence interval on tieredImageNet dataset .

| Model | Backbone | Params | 1-shot | 5-shot |
|---|---|---|---|---|
| MAML (Finn et al., 2017) | ConvNet-4 | 0.11M | $51.67_{\pm1.81}$ | $70.30_{\pm0.08}$ |
| Prototypical-Net (Snell et al., 2017) | ConvNet-4 | 0.11M | $53.31_{\pm0.89}$ | $72.69_{\pm0.74}$ |
| Relation-Net (Sung et al., 2018) | ConvNet-4 | 0.11M | $54.48_{\pm0.93}$ | $71.32_{\pm0.78}$ |
| Simple CNAPS (Bateni et al., 2020) | ResNet-18 | 11M | $63.00_{\pm1.00}$ | $80.00_{\pm0.80}$ |
| LEO-trainval (Rusu et al., 2018) | ResNet-12 | 7.99M | $66.58_{\pm0.70}$ | $85.55_{\pm0.48}$ |
| Shot-Free (Ravichandran et al., 2019) | ResNet-12 | 7.99M | $63.52_{\pm n/a}$ | $82.59_{\pm n/a}$ |
| Fine-tuning (Dhillon et al., 2019) | ResNet-12 | 7.99M | $68.07_{\pm0.26}$ | $83.74_{\pm0.18}$ |
| MetaOptNet (Lee et al., 2019) | ResNet-12 | 12.42M | $65.99_{\pm0.72}$ | $81.56_{\pm0.53}$ |
| RFS-distill (Tian et al., 2020) | ResNet-12 | 13.55M | $71.52_{\pm0.69}$ | $86.03_{\pm0.49}$ |
| EPNet (Rodríguez et al., 2020) | ResNet-12 | 7.99M | $72.60_{\pm0.91}$ | $85.69_{\pm0.65}$ |
| Meta-Baseline (Chen et al., 2021) | ResNet-12 | 7.99M | $68.62_{\pm0.27}$ | $83.74_{\pm0.18}$ |
| IE-distill[1] (Rizve et al., 2021) | ResNet-12 | 13.55M | $72.21_{\pm0.90}$ | $87.08_{\pm0.58}$ |
| **TAS-simple (ours)** | **ResNet-12** | **7.99M** | **$71.98_{\pm0.39}$** | **$86.58_{\pm0.46}$** |
| **TAS-distill (ours)** | **ResNet-12** | **7.99M** | **$72.81_{\pm0.48}$** | **$87.21_{\pm0.52}$** |

[1] performs with wide-layer ResNet-12 with Dropblock as a regularizer

## 6  CONCLUSIONS

A task affinity score, which is non-commutative and invariant to the permutation of labels is introduced in this paper. The application of this affinity score in the few-shot learning is investigated. In particular, the task affinity score is applied to identify the related training classes, which are used for episodically fine-tuning a few-shot model. This approach helps to improve the performance of the few-shot classifier on various benchmarks. Overall, the task affinity score shows the importance of selecting the relevant data for training the neural networks with limited data and can be applied to other applications in machine learning.

ACKNOWLEDGMENTS

This work was supported in part by the Army Research Office grant No. W911NF-15-1-0479.

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

## A  APPENDIX

Here, we fist present the proof of the Theorem 1, and then we provide more experimental results on CIFAR-FS and FC-100 datasets. Additionally, we present the ablation study to show the efficacy of our proposed task affinity score. Moreover, we also discuss about the computation complexity of our approach, and the future works.

### A.1  PROOF OF THEOREM 1

In the proof of Theorem 1, we invoke the Polyak & Juditsky (1992) theorem on the convergence of the average sequence of estimation in different epochs from the SGD algorithm. Since the original form of this theorem has focused on the strongly convex functions, we have assumed that the objective function in the training of the approximation network is strongly convex. However, there are some recent works that try to generalize Polyak & Juditsky (1992) theorem for the convex or even some non-convex functions in an (non)-asymptotic way Gadat & Panloup (2017). Here, we apply only on the asymptotic version of the theorem proposed originally by Polyak & Juditsky (1992). We first recall the definition of the strongly convex function.

**Definition 4** (Strongly Convex Function). *A differentiable function $f : \mathbb{R}^n \to \mathbb{R}$ is strongly convex if for all $x, y \in \mathbb{R}^n$ and some $\mu > 0$, $f$ satisfies the following inequality:*

$$f(y) \geq f(x) + \nabla(f)^T(y - x) + \mu||y - x||_2^2. \tag{6}$$

**Theorem 1.** *Let $X_A$ be the dataset for the task $T_A$ with the objective function $L$, and $X_B$ be the dataset for the task $T_B$. Assume $X_A$ and $X_B$ have the same distribution. Consider an $\varepsilon$-approximation network $N_\theta$ for the pair task-dataset $(T_A, X_A)$ with the objective functions $L$ and with the SGD algorithm to result weights $\theta_t$ at time $t$. Assume that the function $L$ is strongly convex, and its 3rd-order continuous derivative exists and bounded. Let the noisy gradient function in training $N_\theta$ network using SGD algorithm be given by: $g(\theta_t, \epsilon_t) = \nabla L(\theta_t) + \epsilon_t$, where $\theta_t$ is the estimation of the weights for network $N$ at time $t$, and $\nabla L(\theta_t)$ is the true gradient at $\theta_t$. Assume that $\epsilon_t$ satisfies $\mathbb{E}[\epsilon_t | \epsilon_0, ..., \epsilon_{t-1}] = 0$, and satisfies $s = \lim_{t \to \infty} \left|\left|[\epsilon_t \epsilon_t^T | \epsilon_0, \ldots, \epsilon_{t-1}]\right|\right|_\infty < \infty$ almost surely (a.s.). Then the task affinity score between $T_A$ and $T_B$ computed on the average of estimated weights up to the current time $t$ converges to a constant as $t \to \infty$. That is,*

$$s_t = \frac{1}{\sqrt{2}} \left\| \bar{F}_{A_t}^{1/2} - \bar{F}_{B_t}^{1/2} \right\|_F \xrightarrow{\mathcal{D}} \frac{1}{\sqrt{2}} \left\| F_A^{*1/2} - F_B^{*1/2} \right\|_F, \tag{7}$$

*where $\bar{F}_{A_t} = F(\bar{\theta}_t, X_A^{query})$, $\bar{F}_{B_t} = F(\bar{\theta}_t, X_B^{support})$ with $\bar{\theta}_t = \frac{1}{t}\sum_t \theta_t$. Moreover, $F_A^* = F(\theta^*, X_A^{query})$ and $F_B^* = F(\theta^*, X_B^{support})$ denote the Fisher Information matrices computed using the optimum approximation network (i.e., a network with the global minimum weights, $\theta^*$).*

***Proof of Theorem 1.*** Consider task $T_A$ as the source task and task $T_B$ as the target task. Let $\theta_t$ be the set of weights at time $t$ from the $\varepsilon$-approximation network $N$ trained using dataset

$X_A^{support}$ with the objective functions $L$. Since the objective function $L$ is strongly convex, the set of weights $\theta$ will obtain the optimum solutions $\theta^*$ after training a certain number of epochs with stochastic gradient descend. By the assumption on the conditional mean of the noisy gradient function and the assumption on $S$, the conditional covariance matrix is finite as well, i.e., $C = \lim_{t\to\infty} \mathbb{E}[\epsilon_t \epsilon_t^T | \epsilon_0, \ldots, \epsilon_{t-1}] < \infty$; hence, we can invoke the following result due to Polyak et al. Polyak & Juditsky (1992):

$$\sqrt{t}(\bar{\theta}_t - \theta^*) \xrightarrow{\mathcal{D}} \mathcal{N}\Big(0, \mathbf{H}\big(L(\theta^*)\big)^{-1} C \mathbf{H}^T \big(L(\theta^*)\big)^{-1}\Big), \tag{8}$$

as $t \to \infty$. Here, $\mathbf{H}$ is Hessian matrix, $\theta^*$ is the global minimum of the loss function, and $\bar{\theta}_t = \frac{1}{t}\sum_t \theta_t$. In other words, for the network $N$, $\sqrt{t}(\bar{\theta}_t - \theta^*)$ is asymptotically normal random vector:

$$\sqrt{t}(\bar{\theta}_t - \theta^*) \xrightarrow{\mathcal{D}} \mathcal{N}(0, \Sigma), \tag{9}$$

where $\Sigma = \mathbf{H}\big(L(\theta^*)\big)^{-1} C \mathbf{H}^T \big(L(\theta^*)\big)^{-1}$. For a given dataset $X$, the Fisher Information $F_X(\theta)$ is a continuous and differentiable function of $\theta$, and it is also a positive definite matrix; thus, $F_X(\theta)^{1/2}$ is well-defined. Now, by applying the Delta method to Equation (9) with dataset $X_A$ of task $T_A$, we have:

$$(\bar{F}_{At}^{1/2} - F_A^{*\,1/2}) \xrightarrow{\mathcal{D}} \mathcal{N}\big(0, \tfrac{1}{t}\Sigma_A\big), \tag{10}$$

where $\bar{F}_{At} = F_{X_A}(\bar{\theta}_t)$, $\bar{F}_A^* = F_{X_A}(\theta^*)$, and the covariance matrix is given by $\Sigma_A = \mathbf{J}_\theta\Big(\mathbf{vec}\big(F_{X_A}(\theta^*)^{1/2}\big)\Big) \Sigma \mathbf{J}_\theta\Big(\mathbf{vec}\big(F_{X_A}(\theta^*)^{1/2}\big)\Big)^T$. Here, $\mathbf{vec}()$ is the vectorization operator, $\theta^*$ is a $n \times 1$ vector of the optimum parameters, $F_{X_A}(\theta^*)$ is a $n \times n$ Matrix evaluated at the minimum using dataset $X_A$, and $\mathbf{J}_\theta(\mathbf{vec}(F_{X_A}(\theta^*)^{1/2}))$ is a $n^2 \times n$ Jacobian matrix of the Fisher Information Matrix. Likewise, from Equation (9) with dataset $X_B$ of task $T_B$, we have:

$$(\bar{F}_{Bt}^{1/2} - F_B^{*\,1/2}) \xrightarrow{\mathcal{D}} \mathcal{N}\big(0, \tfrac{1}{t}\Sigma_B\big), \tag{11}$$

where $\bar{F}_{Bt} = F_{X_B}(\bar{\theta}_t)$, $\bar{F}_B^* = F_{X_B}(\theta^*)$, and the covariance matrix is given by $\Sigma_B = \mathbf{J}_\theta\Big(\mathbf{vec}\big(F_{X_B}(\theta_B^*)^{1/2}\big)\Big) \mathbf{J}_\theta\Big(\mathbf{vec}\big(F_{X_B}(\theta_B^*)^{1/2}\big)\Big)^T$. From Equation (10) and (11), we obtain:

$$(\bar{F}_{At}^{1/2} - \bar{F}_{Bt}^{1/2}) \xrightarrow{\mathcal{D}} \mathcal{N}\Big(\mu, V\Big), \tag{12}$$

where $\mu = \big(F_A^{*\,1/2} - F_B^{*\,1/2}\big)$ and $V = \frac{1}{t}(\Sigma_A + \Sigma_B)$. Since $(\bar{F}_{At}^{1/2} - \bar{F}_{Bt}^{1/2}) - (F_A^{*\,1/2} - F_B^{*\,1/2})$ is asymptotically normal with the covariance goes to zero as $t$ approaches infinity, all of the entries go to zero, we conclude that:

$$s_t = \frac{1}{\sqrt{2}} \left\| \bar{F}_{At}^{1/2} - \bar{F}_{Bt}^{1/2} \right\|_F \xrightarrow{\mathcal{D}} \frac{1}{\sqrt{2}} \left\| F_A^{*\,1/2} - F_B^{*\,1/2} \right\|_F. \tag{13}$$

$\square$

## A.2 EXPERIMENTS ON CIFAR-FS AND FC-100

Here, we conduct the experiments on the CIFAR-FS and the FC-100 datasets. The CIFAR-FS dataset (Bertinetto et al., 2018) is derived from the original CIFAR-100 dataset (Krizhevsky et al., 2009) by splitting 100 classes into 64 training classes, 16 validation classes, and 20 testing classes. The FC-100 dataset (Oreshkin et al., 2018) is also a subset of the CIFAR-100 and consists of 60 training classes, 20 validation classes, and 20 testing classes. The size of data sample in both of these datasets is $32 \times 32$.

Similar to previous experiments on miniImageNet and tieredImageNet, we train the ResNet-12 for 100 epochs with batch size of 64, and the learning rate decaying at epoch 60. Next, we compare our proposed few-shot approach with other state-of-the-art methods on CIFAR-FS and FC-100 datasets. The results in Table 3, and Table 4 indicates the competitiveness of our method in both datasets. Our approach with standard ResNet-12 is comparable to IE-distill (Rizve et al., 2021) and outperforms RFS-distill (Tian et al., 2020) while having a significantly smaller classifier model in term of the number of parameters.

Table 3: Comparison of the accuracy against state-of-the art methods for 5-way 1-shot and 5-way 5-shot classification with 95% confidence interval on CIFAR-FS dataset.

| | | | CIFAR-FS | |
|---|---|---|---|---|
| Model | Backbone | Params | 1-shot | 5-shot |
| MAML (Finn et al., 2017) | ConvNet-4 | 0.11M | $58.90_{\pm 1.90}$ | $71.50_{\pm 1.00}$ |
| Prototypical-Net (Snell et al., 2017) | ConvNet-4 | 0.11M | $55.50_{\pm 0.70}$ | $72.00_{\pm 0.60}$ |
| Relation-Net (Sung et al., 2018) | ConvNet-4 | 0.11M | $55.00_{\pm 1.00}$ | $69.30_{\pm 0.80}$ |
| Prototypical-Net (Snell et al., 2017) | ResNet-12 | 7.99M | $72.20_{\pm 0.70}$ | $83.50_{\pm 0.50}$ |
| Shot-Free (Ravichandran et al., 2019) | ResNet-12 | 7.99M | $69.20_{\pm n/a}$ | $84.70_{\pm n/a}$ |
| TEWAM (Qiao et al., 2019) | ResNet-12 | 7.99M | $70.40_{\pm n/a}$ | $81.30_{\pm n/a}$ |
| MetaOptNet (Lee et al., 2019) | ResNet-12 | 12.42M | $72.60_{\pm 0.70}$ | $84.30_{\pm 0.50}$ |
| RFS-simple (Tian et al., 2020) | ResNet-12 | 13.55M | $71.50_{\pm 0.80}$ | $86.00_{\pm 0.50}$ |
| RFS-distill (Tian et al., 2020) | ResNet-12 | 13.55M | $73.90_{\pm 0.80}$ | $86.90_{\pm 0.50}$ |
| IE-distill[1] (Rizve et al., 2021) | ResNet-12 | 9.13M | $75.46_{\pm 0.86}$ | $88.67_{\pm 0.58}$ |
| **TAS-simple (ours)** | **ResNet-12** | **7.99M** | $\mathbf{73.47_{\pm 0.42}}$ | $\mathbf{86.82_{\pm 0.49}}$ |
| **TAS-distill (ours)** | **ResNet-12** | **7.99M** | $\mathbf{74.02_{\pm 0.55}}$ | $\mathbf{87.65_{\pm 0.58}}$ |
| **TAS-distill[2] (ours)** | **ResNet-12** | **12.47M** | $\mathbf{75.56_{\pm 0.62}}$ | $\mathbf{88.95_{\pm 0.65}}$ |

[1] performs with standard ResNet-12 with Dropblock as a regularizer, [2] performs with wide-layer ResNet-12

Table 4: Comparison of the accuracy against state-of-the art methods for 5-way 1-shot and 5-way 5-shot classification with 95% confidence interval on FC-100 dataset.

| | | | FC-100 | |
|---|---|---|---|---|
| Model | Backbone | Params | 1-shot | 5-shot |
| Prototypical-Net (Snell et al., 2017) | ConvNet-4 | 0.11M | $35.30_{\pm 0.60}$ | $48.60_{\pm 0.60}$ |
| Prototypical-Net (Snell et al., 2017) | ResNet-12 | 7.99M | $37.50_{\pm 0.60}$ | $52.50_{\pm 0.60}$ |
| TADAM (Oreshkin et al., 2018) | ResNet-12 | 7.99M | $40.10_{\pm 0.40}$ | $56.10_{\pm 0.40}$ |
| MetaOptNet (Lee et al., 2019) | ResNet-12 | 12.42M | $41.10_{\pm 0.60}$ | $55.50_{\pm 0.60}$ |
| RFS-simple (Tian et al., 2020) | ResNet-12 | 13.55M | $42.60_{\pm 0.70}$ | $59.10_{\pm 0.60}$ |
| RFS-distill (Tian et al., 2020) | ResNet-12 | 13.55M | $44.60_{\pm 0.70}$ | $60.90_{\pm 0.60}$ |
| IE-distill[1] (Rizve et al., 2021) | ResNet-12 | 9.13M | $44.65_{\pm 0.77}$ | $61.24_{\pm 0.75}$ |
| **TAS-simple (ours)** | **ResNet-12** | **7.99M** | $\mathbf{43.10_{\pm 0.67}}$ | $\mathbf{60.65_{\pm 0.62}}$ |
| **TAS-distill (ours)** | **ResNet-12** | **7.99M** | $\mathbf{44.62_{\pm 0.70}}$ | $\mathbf{61.46_{\pm 0.65}}$ |

[1] performs with standard ResNet-12 with Dropblock as a regularizer

## A.3 ABLATION STUDY

First, we show the stability of the TAS by applying our distance on various classification tasks in CIFAR-10, CIFAR-100, ImageNet datasets using ResNet-18, and VGG-16 as the backbone for the $\varepsilon$-approximation networks. For each dataset, we define 4 classification tasks, all of which are variations of the full class classification task. For each task, we consider a balanced training dataset. That is, except for the classification tasks with all the labels, only a subset of the original training dataset is used such that the number of training samples across all the class labels to be equal. Additionally, we use 3 different $\varepsilon$ values for the $\varepsilon$-approximation networks, ranging from 0.2 (good) to 0.6 (bad). To make sure that our results are statistically significant, we run our experiments 10 times with each of the $\varepsilon$-approximation networks being initialized with a different random seed each time and report the mean and the standard deviation of the computed distance.

In CIFAR-10, we define 4 tasks as follow:

- Task 0 is a binary classification of indicating 3 objects: automobile, cat, ship (i.e., the goal is to decide if the given input image consists of one of these three objects or not).
- Task 1 is a binary classification of indicating 3 objects: cat, ship, truck.
- Task 2 is a 4-class classification with labels bird, frog, horse, and anything else.
- Task 3 is the standard 10 objects classification.

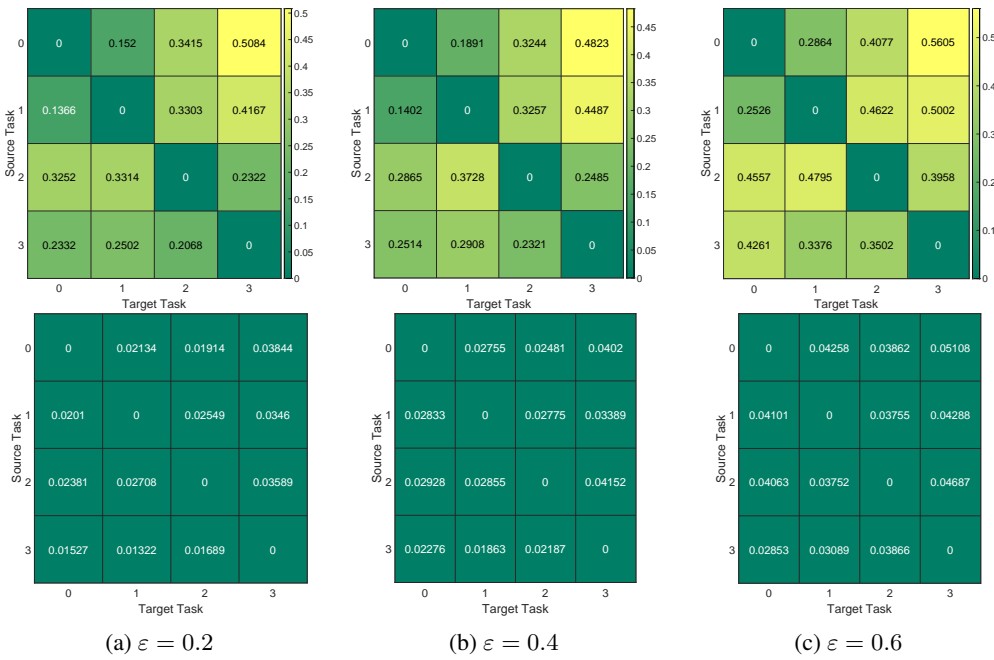

Figure 3: Distance from source tasks to the target tasks on CIFAR-10 using ResNet-18 backbone. The top row shows the mean values and the bottom row denotes the standard deviation of distances between classification tasks over 10 different trials.

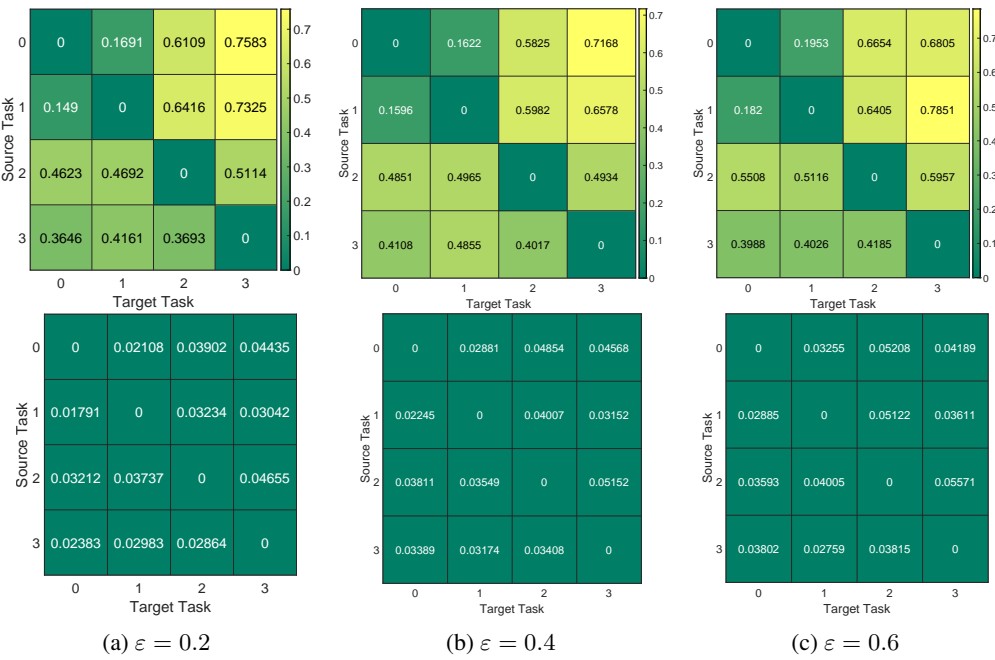

Figure 4: Distance from source tasks to the target tasks on CIFAR-10 using VGG-16 backbone. The top row shows the mean values and the bottom row denotes the standard deviation of distances between classification tasks over 10 different trials.

In Figure 3, the mean and standard deviation of the TAS between CIFAR-10 tasks over 10 trial runs, using 3 different $\varepsilon$ values for the ResNet-18 approximation networks. For $\varepsilon = 0.2$, the approximation network's performance on the corresponding task's test data is at least $80\%$ and it is considered to be a good representation. As shown in Figure 3(a), the standard deviation of TAS is relatively

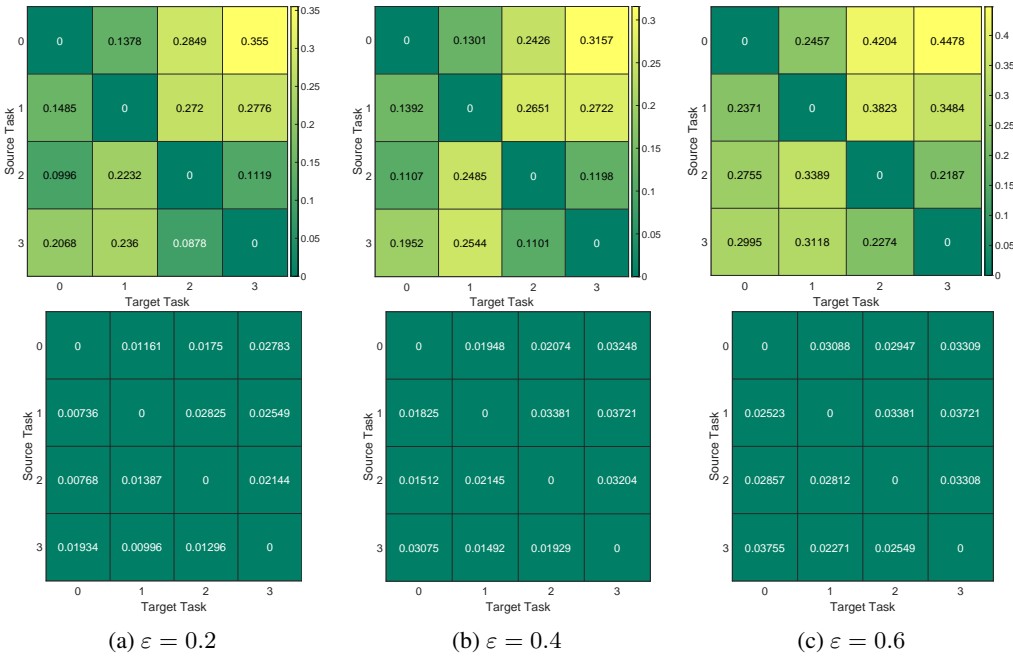

(a) $\varepsilon = 0.2$        (b) $\varepsilon = 0.4$        (c) $\varepsilon = 0.6$

Figure 5: Distance from source tasks to the target tasks on CIFAR-100 using ResNet-18 backbone. The top row shows the mean values and the bottom row denotes the standard deviation of distances between classification tasks over 10 different trials.

small and the TAS is stable (or consistent) regardless of the network's initialization. Thus, we can easily identify the closest tasks to target tasks. The result for $\varepsilon = 0.4$, shown in Figure 3(b), the order of tasks remains the same for most cases, however, the standard deviations are much larger than the previous case. Lastly, Figure 3(c) shows the results for $\varepsilon = 0.6$. In this setup, TAS between tasks fluctuates widely and it is considered to be unreliable. Consequently, if the TAS between a pair of tasks is computed only once, the order of the closest tasks to the target task would easily be incorrect. Similarly, we conduct the experiment in CIFAR-10 using VGG-16 as the backbone. In Figure 4, we observe similar trend as the approximation network with smaller $\varepsilon$ value will obtain the more stable and consistent task distances.

In CIFAR-100, which consisting of 100 objects equally distributed in 20 sub-classes, each sub-class has 5 object, we define 4 tasks as follow:

- Task 0 is a binary classification of detecting an object that belongs to vehicles 1 and 2 sub-classes or not (i.e., the goal is to decide if the given input image consists of one of these 10 vehicles or not).
- Task 1 is a binary classification of detecting an object that belongs to household furniture and household devices or not.
- Task 2 is a multi-classification with 11 labels defined on vehicles 1, vehicles 2, and anything else.
- Task 3 is a multi-classification with the 21-labels in vehicles 1, vehicles 2, household furniture, household devices, and anything else.

In Figure 5, the mean and standard deviation of the TAS between CIFAR-100 tasks over 10 trial runs, using 3 different $\varepsilon$ values for the ResNet-18 approximation networks. For $\varepsilon = 0.2$, the approximation network's performance on the corresponding task's test data is at least $80\%$ and it is considered to be a good representation. As shown in Figure 5(a), TAS is stable regardless of the network initialization. As $\varepsilon$ increases from $0.2$ to $0.6$, we observe the significant rise in standard deviation. For $\varepsilon = 0.6$, the TAS results are widely fluctuating and unreliable. In particular, consider Task 0 as the incoming task, and Task $1, 2, 3$ are the base tasks. From the approximation network $\varepsilon = 0.2$ in Figure 5(a), the closest task to Task 0 is Task 2. However, the approximation network

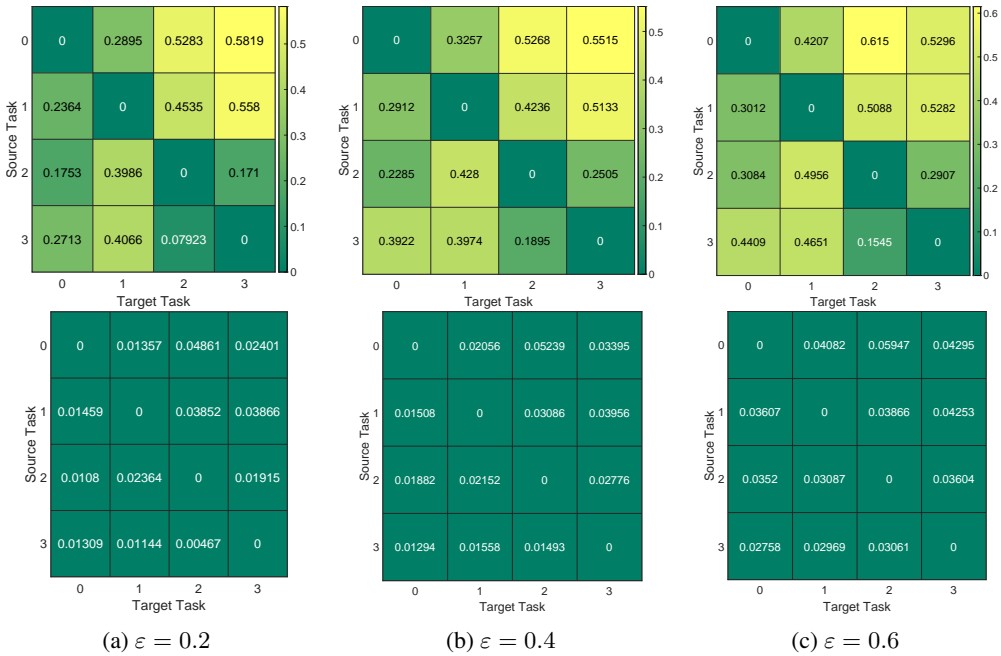

Figure 6: Distance from source tasks to the target tasks on CIFAR-100 using VGG-16 backbone. The top row shows the mean values and the bottom row denotes the standard deviation of distances between classification tasks over 10 different trials.

$\varepsilon = 0.6$ in Figure 5(c) identifies Task 1 as the closest task of Task 0. Thus, in order to achieve a meaningful TAS between tasks, we first need to make sure that we represent the tasks correctly by the approximation networks. If the approximation network is not good in term of performance (i.e., $\varepsilon$ is large), then, that network is not a good representation for the task. Similarly, we conduct the experiment in CIFAR-100 using VGG-16 as the backbone. In Figure 6, we observe similar trend as the approximation network with smaller $\varepsilon$ value will obtain the more stable and consistent task distances.

In ImageNet, we define four 10-class classification tasks in ImageNet dataset. For each class, we consider 800 for training and 200 for the test samples. The list of 10 classes for each task is described below:

- Task 0 includes tench, English springer, cassette player, chain saw, church, French horn, garbage truck, gas pump, golf ball, parachute.

- Task 1 is similar to Task 0; however, instead of 3 labels of tench, golf ball, and parachute, it has samples from the grey whale, volleyball, umbrella classes.

- In Task 2, we also replace 5 labels of grey whale, cassette player, chain saw, volleyball, umbrella in Task 0 with another 5 labels given by platypus, laptop, lawnmower, baseball, cowboy hat.

- Task 3 includes analog clock, candle, sweatshirt, birdhouse, ping-pong ball, hotdog, pizza, school bus, iPod, beaver.

In Figure 7 and Figure 8, we demonstrate the mean and standard deviation of the TAS between 4 ImageNet tasks over 10 runs using ResNet-18 and VGG-16 as the backbone. For each backbone, we conduct 3 different $\varepsilon$ values and report results as the above experiments. Similar to the CIFAR-10 and CIFAR-100, we observe that the approximation network with better performance (or smaller $\varepsilon$ value) will results more stable TAS. Additionally, regardless of the architecture backbone, the trend of distance between these tasks is consistent across all experiments.

Next, we want to analyze the effectiveness of our approach of choosing the most related data classes by performing several few-shot learning experiments with different number of related classes. Ad-

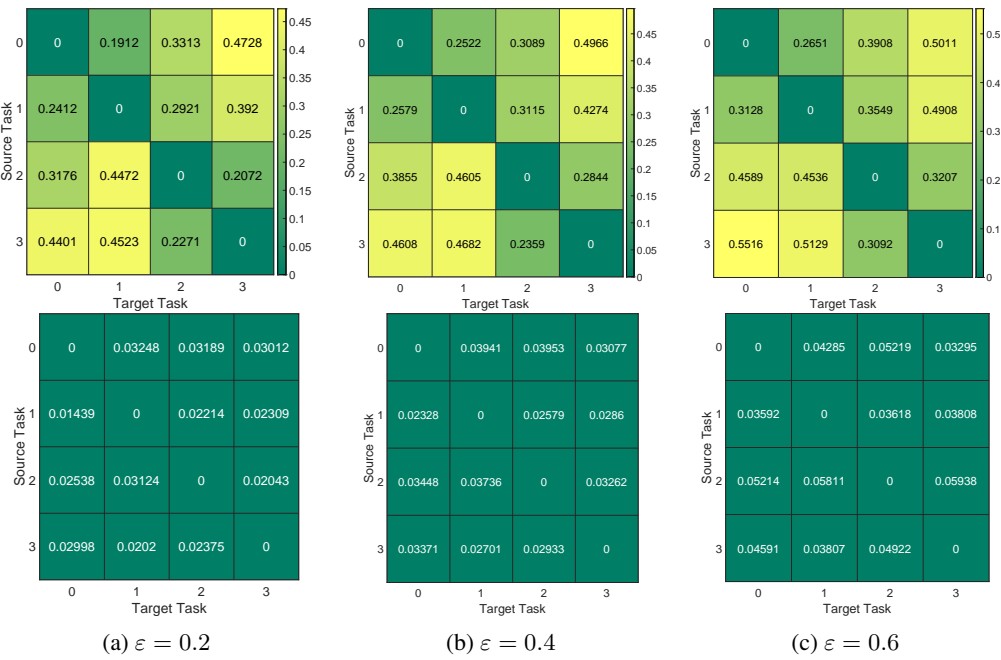

Figure 7: Distance from source tasks to the target tasks on ImageNet using ResNet-18 backbone. The top row shows the mean values and the bottom row denotes the standard deviation of distances between classification tasks over 10 different trials.

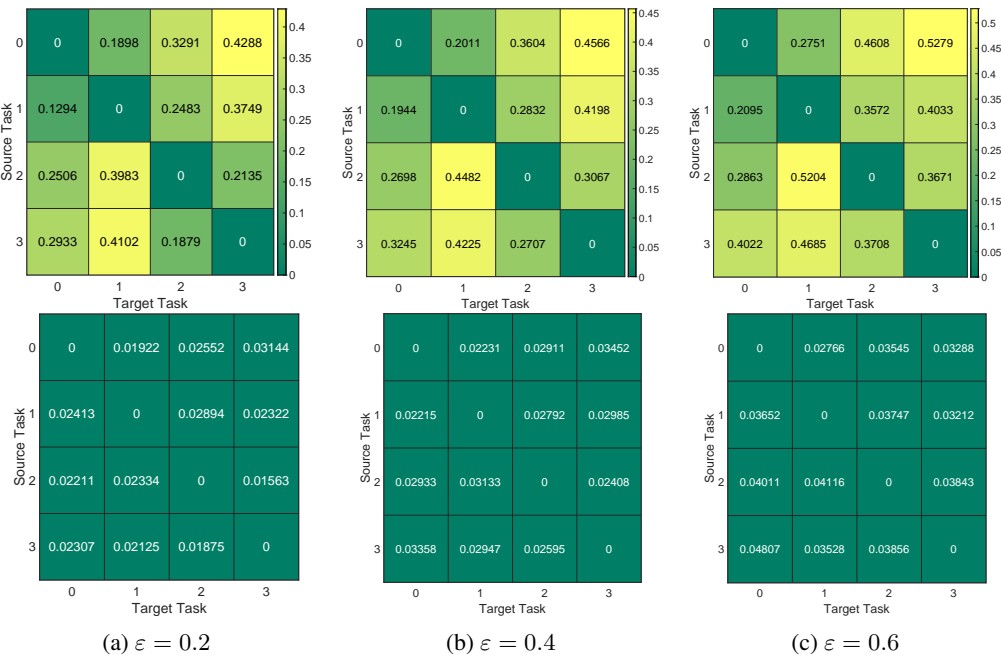

Figure 8: Distance from source tasks to the target tasks on ImageNet using VGG-16 backbone. The top row shows the mean values and the bottom row denotes the standard deviation of distances between classification tasks over 10 different trials.

ditionally, we conduct the few-shot experiment with non-related classes and randomized classes to show the efficacy of our proposed method. Table 5 shows the performances of our proposed few-shot method with different settings. We achieve the best performance with 49 related tasks in both 5-way 1-shot and 5-way 5-shot classification. We observe a drop in performance when the number

Table 5: The performance of TAS with different settings for 5-way 1-shot and 5-way 5-shot classification with 95% confidence interval on miniImageNet dataset.

| Model | Params | miniImageNet | |
|---|---|---|---|
| | | 1-shot | 5-shot |
| TAS-simple with top-54 related classes | 7.99M | $63.54_{\pm0.49}$ | $80.92_{\pm0.55}$ |
| **TAS-simple with top-49 related classes** | **7.99M** | **$64.71_{\pm0.43}$** | **$82.08_{\pm0.45}$** |
| TAS-simple with top-38 related classes | 7.99M | $63.92_{\pm0.47}$ | $81.29_{\pm0.50}$ |
| TAS-simple with top-32 related classes | 7.99M | $63.35_{\pm0.55}$ | $80.67_{\pm0.53}$ |
| TAS-simple with top-29 non-related classes | 7.99M | $61.82_{\pm0.44}$ | $78.14_{\pm0.48}$ |
| TAS-simple with 32 random classes | 7.99M | $62.57_{\pm0.45}$ | $78.96_{\pm0.44}$ |

Table 6: The performance TAS with different settings for 5-way 1-shot and 5-way 5-shot classification with 95% confidence interval on tieredImageNet dataset.

| Model | Params | miniImageNet | |
|---|---|---|---|
| | | 1-shot | 5-shot |
| TAS-simple with top-318 related classes | 7.99M | $71.02_{\pm0.42}$ | $85.26_{\pm0.48}$ |
| **TAS-simple with top-293 related classes** | **7.99M** | **$71.98_{\pm0.39}$** | **$86.58_{\pm0.46}$** |
| TAS-simple with top-249 related classes | 7.99M | $70.89_{\pm0.45}$ | $85.65_{\pm0.52}$ |
| TAS-simple with top-211 related classes | 7.99M | $70.11_{\pm0.45}$ | $84.86_{\pm0.50}$ |
| TAS-simple with top-223 non-related classes | 7.99M | $67.33_{\pm0.50}$ | $83.24_{\pm0.52}$ |
| TAS-simple with 211 random classes | 7.99M | $68.04_{\pm0.46}$ | $83.55_{\pm0.52}$ |

of related classes increases to $54$ classes. As we reduce the number of related classes from $49$ to $32$ classes, the overall performance of our model reduces significantly in both 1-shot and 5-shot. As the result, selecting the optimal number of relevant classes is crucial to the overall performance of our few-shot approach. Additionally, we observe a drop in performance when the model utilizing $29$ non-related classes as well as $32$ random classes. Therefore, we show that using a relevant classes can boost the performance of the few-shot model. Moreover, we conduct the similar analysis for the tieredImageNet dataset, in which the results are shown in Table 6. Here, we observe the same trend as in previous experiments. Briefly, by selecting the relevant classes, we consistently achieve higher performances when compare with selecting non-relevant or randomized classes.

## A.4 COMPUTATION COMPLEXITY

Below are the complexity of 3 phases in our approach:

- Phase 1 consists of 2 operations: (i) Train Whole-Classification net. (ii) Feature extraction. These 2 operations is executed only once, and the complexity is constant.

- Phase 2 consists of 3 operations: (iii) Matching labels (iv) Constructing approx. net. (v) Fisher Information matrix. The operation (iii) is based on the Hungarian algorithm and the complexity is $O(n^3)$, where n is the number of tasks. For each base task, (iv) is done once, and the complexity of finding FIM in (v) is linear with respect to the number of parameters due to the fact that we only consider the diagonal entries of the matrix. Note that all the tasks are few-shot and only consists of few data samples. These operations can be effectively implemented in GPU.

- Phase 3 is (vi) Episodic fine-tuning, which is often used in few-shot approach [R3, R4]. Here, we use the k-NN as the clustering algorithm. The complexity of k-NN is linear with respect to the number of samples.

## A.5 FUTURE WORKS

In this paper, we propose a novel task similarity measure that inherently asymmetric and invariant to label permutation, called Task Affinity Score (TAS). Next, we introduce the TAS measure to the

few-shot learning approach and achieve significant improvements over the current state-of-the-art methods. In future work, we would like to extend the applications of TAS to other few-shot learning frameworks, and also to other meta-learning areas, e.g., multi-task learning, continual learning. Additionally, we wish to establish theoretical justification for the TAS measure in convex or non-convex cases.

