# OpenReview forum: "Task Affinity with Maximum Bipartite Matching in Few-Shot Learning"
_ICLR.cc/2022/Conference — ICLR 2022 Poster_

### Official Review · Reviewer_Y7LN · 2021-11-02

**Correctness:** 3
**Technical Novelty And Significance:** 3
**Empirical Novelty And Significance:** 2
**Recommendation:** 6
**Confidence:** 4

**Main Review:**

pros:
The paper is well-written and neatly presented.
The idea that leveraging task affinity score to find most relevant tasks for better few-shot learning is impressive, and seems to be technically sound.

cons:
	The contributions are not well highlighted.
	There is no ablation study in this paper, the experimental results are insufficient to validate the idea and many detailed experiments are not provided.
	The main idea and the motivation are easy to follow, however some details of the proposed model are still not well specified.

Detailed comments and questions:
	1. In Definition 1, what is the meaning of epsilon-approximate network? How do you determine the value of epsilon, does it have a significant impact on the performance of the model?
	2. In Definition 3, why use task Ta's query data to compute the Fisher information matrix instead of using support data?
	3. It is mentioned in this paper that the classifier model is fine-tuned only with the labels of the related-training set, the complexity of training of the few-shot model is reduced. However, from Algorithm 1, the relevant task selection seems quite time consuming. Has the author considered the computational complexity of the model?
	4. In 1-shot setting, the class centroids of the target task are the sample itself. Will the closest source task calculated in this case be biased due to the randomness of the sample selection?
	5. In Section 4.2.2, the author says that top-R scores and their corresponding classes are finally selected. Then Algorithm 1 selects the task with the minimum TAS. Which method does the author use?
	6. How to use the source task fine-tuning model, which the authors did not explain? For example, in the miniImageNet, 2000 source tasks are randomly selected, then the model is fine-tuned on each source task to calculate TAS to the target task. Is this process time-consuming?
	7. In the tieredImageNet, why only 120 labels are selected for each source task when the test set of this dataset has 160 labels?
	8. In Table 1 and Table 2, why use different backbone to compare IE-Distill methods respectively? miniImageNet uses standard resnet-12 while tieredImageNet uses wide-layer resnet-12.
	9. What is the baseline of the paper? Is the performance gain from TAS-Simple to TAS-Distill derived from knowledge distillation or the method proposed by the authors?


**Summary Of The Paper:**

The authors propose a task affinity score based on maximum bipartite matching algorithm and Fisher information matrix. And then utilize this score to find the closest training data labels to the test data and leverage the discovered relevant data for episodically fine-tuning the few-shot model. Experimental results on few-shot learning setting achieve the state-of-the-art performance on four widely-used benchmarks.

**Summary Of The Review:**

The overall idea of using task affinity score for few-shot learning is impressive. However, I think this submission is incomplete and lacks a lot of technical details, which makes the paper not easy to be understood. Meanwhile, the paper also has not conducted any ablation study to show the real empirical improvement of the proposed strategies and modules. Moreover, although the method proposed by the author does not introduce additional model parameters, I am quite concerned that the computation cost of the fine-tuning source tasks during the task affinity score calculation.

---

> ### Author Response · Authors · 2021-11-12
> **Official Response to Reviewer Y7LN**
>
> Thank you very much for your very important and constructive feedback. Here we want to address your concerns and below are our answers:
>
> 1. The ϵ-approximation is a well-trained neural network that we used to represent a task. The value of epsilon indicates that the network can perform the corresponding task with the accuracy P ≥ 1 − ϵ. If ϵ is too large, the approximation model would perform poorly on the task, and would not be able to represent the task correctly. In the appendix, we include experiments showing that using a bad approximation network (i.e., large ϵ) can affect the consistency of the TAS. By running the experiment 10 times with different initialization, the TAS’s variance of the bad approx. network is much larger than the TAS’s variance from the good approx. network. For example, in Figure 4, consider Task 0 as the incoming task, and Task 1, 2, 3 are the base tasks. From the approximation network ϵ= 0.2, the closest task to Task 0 is Task 2.  However, the approximation network ϵ= 0.6 identifies Task 1 as the closest task of Task 0. Please see the ablation study (pages 15-18) in the appendix for more detail.
>
> 2. The approximation network is trained using the support set of the base task. Then, we want to compute 2 Fisher Information matrices, i.e., first using the query set of the base task, second using the support set of the test task. The reason for us to use the query set of the base task is that the network hasn’t been exposed to this part of the data from the base task, and we want to see whether the network can perform well on this query set while comparing to the network’s performance on the support set of the test task (only the support set of the test task is available during training, and the network also hasn’t exposed to these data samples either).
>
> 3. Thank you for your suggestion. We agree that the complexity of the approach should be mentioned in the paper, though other papers often overlook this part. Below are the complexity of 3 phases in our approach, which we also included in the appendix section (on pages 17-18):
> * Phase 1 consists of 2 operations: (i) Train Whole-Classification net. (ii) Feature extraction. These 2 operations are executed only once, and the complexity is constant.
> * Phase 2 consists of 3 operations: (iii) Matching labels (iv) Constructing approx. net. (v) Fisher Information matrix. The operation (iii) is based on the Hungarian algorithm and the complexity is O(n^3), where n is the number of tasks. For each base task, (iv) is done once, and the complexity of finding FIM in (v) is linear with respect to the number of parameters. Note that all the tasks are few-shot and only consist of a few data samples. These operations
> can be effectively implemented in GPU.
> * Phase 3 is (vi) Episodic fine-tuning, which is often used in the few-shot approach [R3, R4]. Here, we use the k-NN as the clustering algorithm. The complexity of k-NN is linear with respect to the number of samples.
>
> 4. In a 1-shot setting, each class in the few-shot test set will only have one sample. The model in our approach is trained to extract the meaningful features, and given that there is 1 sample or multiple samples, we expect that the extracted centroid features are well-represented. Thus, we believe that there is no bias in this setting.
>
> 5. In this paper, we consider a number of closest tasks to help the few-shot learning procedure. In Figure 2, we show that using different numbers of closest tasks will result in different numbers of related labels. Moreover, in the Ablation Study, we discuss the few-shot performance for different choices of related labels. Please refer to pages 17-18 for more detail.
>
> 6. Here we apply the same training scheme as meta-learning approaches. For example, we sample 2000 few-shot tasks in the miniImageNet and train the network using cross-validation loss. The loss at each iteration is the average loss. The training/fine-tuning procedure is not time-consuming due to the fact that all the tasks are few-shot and only contain very few data samples.
>
> 7. That is a typo and the correct number is 160. We have corrected this in the revised version.
>
> 8. IE-Distill approach uses a backbone network with a Dropblock regularizer. Here, we want to show that, in tieredImageNet dataset, we can outperform IE-Distill, which has a wide-ResNet of 13.55M parameters, while using a much smaller model. This advantage occurs in datasets with a large number of classes (e.g., tieredImageNet).
>
> 9. Meta-Baseline (Chen et al., 2021) is a baseline for our approach. TAS-Simple is our whole proposed approach, without the knowledge distillation (KD). TAS-Distill is our method with KD. The performance gain from TAS-Simple to TAS-Distill is derived from KD.

---

### Official Review · Reviewer_UXcu · 2021-11-02

**Correctness:** 3
**Technical Novelty And Significance:** 3
**Empirical Novelty And Significance:** 4
**Recommendation:** 8
**Confidence:** 3

**Main Review:**

Strength:
1. The idea to exploit Fisher Information Matrix for relevance measurement between different tasks is novel.
2. The experimental results are promising.

Weakness:
1. The paper exploits the empirical Fisher Information matrix. The reason for this approximation should be discussed in detail.
2. The authors did not conduct some necessary experiments to validate the effectiveness of TSA, e.g., the authors should prove that models trained with source tasks selected via TSA are consistently better than those trained with randomly sampled source tasks.
3. The proof of TSA holds when the loss function is strictly convex, which is a strong assumption.

**Summary Of The Paper:**

This paper proposes a few shot learning method, which tries to measure the affinity degree between different tasks. Based on the affinity score, the relevant tasks are exploited for training to boost the performance of target tasks. The Task Affinity Score (TSA) is proposed, which is novel  to measure the dependency between different tasks. The reasonability of TSA is also validated with mathematical proof.

**Summary Of The Review:**

This paper exploits the fisher matrix to measure the relevance between different tasks which is an interesting attempt. In addition, the experimental validate the effectiveness of the proposed method.

---

> ### Author Response · Authors · 2021-11-12
> **Official Response to Reviewer UXcu**
>
> Thank you very much for your highly constructive comments and your recommendations. Here we want to address your concerns and below are our answers:
>
> 1. As often defined in meta-learning literature, we define a task by the data and the loss function.
> Next, we train a neural network, called the approximation network, and use it to represent the corresponding task. Since computing the Fisher Information for a task is intractable unless we have an infinite number of data samples, we use the approximation network and compute the empirical Fisher Information Matrix (FIM) for this network for the given data samples. Additionally, to reduce the complexity of computing FIM, we consider only the diagonal entries of the FIM and use them to compute the task affinity score (TAS).
>
> 2. We have conducted experiments using the non-closest tasks in our few-shot learning approach and the results are consistently not as good as the closest tasks. Please see the ablation study in the appendix (pages 15-18) for more detail.
>
> 3. In the theorem section, we want to establish the proof that our task affinity score is well-defined and consistent, regardless of the network initialization. Here, we apply the Polyak Theorem for the strictly convex cases. There are approaches using Polyak Theorem on the convex case, and we will pursue them in our future works (as described in the Future Works Section on page 18).

---

### Official Review · Reviewer_mtpL · 2021-11-08

**Correctness:** 2
**Technical Novelty And Significance:** 2
**Empirical Novelty And Significance:** 3
**Recommendation:** 3
**Confidence:** 3

**Main Review:**

Strong points.
- The authors introduce a new score based on the Fisher Information matrix; its mathematical analysis is also presented.
- The performance is compared with many meta-learning methods in the empirical study.
- The paper is well-organized.

Weak points.
- The proposed few-shot learning scheme seems incremental.
- The advantage of the proposed score (TAS) is not clear.
- A more thorough evaluation is needed to show the effectiveness of the proposed method.

Comments.
1. In my understanding, the main contribution of this study is to incorporate a process of constructing a related-training set via Task Affinity Score (TAS) into a few-shot learning procedure. Another approach to consider the task similarity is to embed tasks into hidden space, as in conditional neural processes (NPs) [R1]. The authors should clarify the advantage of the proposed approach compared with such prior works. Also, it would be helpful to add the empirical evaluation with conditional NPs, etc.

2. In Section 3, the authors present TAS as a general one; however, the few-shot learning scheme is constructed using a specific approach, i.e., Whole-Classification. Is it possible to use TAS for few-shot learning based on another approach?

3. In Tables 1 and 2, the authors demonstrate that the proposed method outperforms many previous meta-learning methods in terms of prediction accuracy. However, this is not enough to convince the effectiveness of the proposed method. I would like to see more evidence of why the proposed method works well. For example, it would be helpful to analyze the related source tasks selected by TAS.

4. As another aspect, the authors would add a discussion on the computation complexity of the proposed method.

[R1] M. Garnelo, D. Rosenbaum, C. Maddison, T. Ramalho, D. Saxton, M. Shana- han, Y. W. Teh, D. Rezende, and S. A. Eslami. Conditional neural processes. In International Conference on Machine Learning, pages 1690–1699, 2018.

**Summary Of The Paper:**

This paper presents a new affinity score based on the Fisher Information matrix from a source to a target task. The authors also develop a few-shot learning procedure based on a pre-trained Whole-Classification network approach. In this procedure, training labels are matched via a maximum matching algorithm, and the target task for training is selected using the proposed affinity score. The effectiveness of the proposed score is demonstrated using benchmark datasets in the problem of few-shot learning.

**Summary Of The Review:**

This paper proposes a new few-shot learning procedure and shows its effectiveness in terms of prediction accuracy. However, the advantage of the proposal should be discussed compared with previous works; the authors should conduct a more thorough evaluation. Accordingly, I vote for rejecting this paper.

---

> ### Author Response · Authors · 2021-11-12
> **Official Response to Reviewer mtpL**
>
> Thank you for your constructive comments. Here are our answers to your above concerns:
> 1. In this paper, we propose a novel task similarity measure, which is asymmetric (or non-commutative) and invariant to label permutation. Next, we introduce a few-shot learning approach that utilizes the relevant data to improve the few-shot model and compare our approach with the state-of-the-art methods and benchmarks in few-shot learning. Although we think [R1, R2] are fundamental pioneering works on conditional neural processes (NPs), we believe that our work is very different, and we didn’t do the comparison because of the following major differences:
> * [R2], which is a variation of using NPs [R1] on meta-learning datasets, e.g., miniImageNet, tieredImageNet, use external data and pretrain the model with the full ImageNet dataset. Additionally, we added the results of using conditional NPs without the extra data in Table 1, 2. As shown in these tables, our method consistently outperforms [R2] in both 1-shot and 5-shot in miniImageNet and tieredImageNet. Please refer to the revised version for more detail.
> * [R1,R2] are based on embedding a task into a hidden space, and the distance between tasks is symmetric. In contrast, we define and represent a task by an approximation network, and our Task Affinity Score (TAS) is inherently asymmetric as task distances should be: having learned a superset of a task, learning a subset task is easier but not the other way around.
> * TAS is inherently invariant to label permutation. For instance, a task of classifying dog (labeled as 0) and cat (labeled as 1) is the same as classifying cat (labeled as 0) and dog (labeled as 1). This is not the case for [R1,R2]. Therefore our method can evaluate not just the relevant tasks, but the relevant labels/classes.
> Briefly, with TAS, we don’t need to collect a group of tasks to perform meta-learning from scratch. We can find the most relevant tasks/labels and borrow their knowledge. This helps booster the values of previous tasks and speed of solving the current problem at hand.
>
> 2. This is a very important comment, and the answer is affirmative. With the main goal of utilizing the relevant data to improve the overall performance, TAS can be applied to other few-shot approaches to identify the most useful data in the training set, given that the task definition is defined based on the data and the loss function. We include the applications of TAS in other few-shot methods to the potential works section in the appendix. Note that, in section 3, we introduce the definition of utilizing the Fisher Information for task similarity. In order to compute the Fisher Information Matrix (FIM) of a task in practice, we need to represent a task by a well-trained approximation network (or the Whole-Classification as often used in few-shot literature [R3,R4]), and then compute the empirical FIM.
>
> 3. Here, we follow a similar protocol from other few-shot learning papers to report the results, which consists of the accuracy, the number of parameters, and the model backbone. In Figure 2, we show the relevant labels to the test tasks in miniImageNet and tieredImageNet. Additionally, we include the ablation study in the appendix, which shows a drop in performance when choosing the non-related classes for fine-tuning. In this study, we also provide empirical experiments to show that TAS is well-defined and consistent, given that we have a good approximation network to represent a task. Please see the appendix (pages 15-18) for more detail.
>
> 4. Thank you for your important suggestion. We included the computational complexity of 3 phases in our approach in the revised version. Please see the appendix section (pages17-18) for more detail.
>
>
> [R2] Bateni, Peyman, Raghav Goyal, Vaden Masrani, Frank Wood, and Leonid Sigal. ”Improved few-shot visual classification.” In Proceedings of the IEEE/CVF Conference on Computer Vision and Pattern Recognition, pp. 14493-14502. 2020.
>
> [R3] Yinbo Chen, Zhuang Liu, Huijuan Xu, Trevor Darrell, and Xiaolong Wang. Meta-baseline: exploring simple meta-learning for few-shot learning. In Proceedings of the IEEE/CVF International Conference on Computer Vision, 2021.
>
> [R4] Yonglong Tian, Yue Wang, Dilip Krishnan, Joshua B Tenenbaum, and Phillip Isola. Rethinking few-shot image classification: a good embedding is all you need? In Computer Vision–ECCV 2020: 16th European Conference, Glasgow, UK, August 23–28, 2020, Proceedings, Part XIV 16, pp. 266–282. Springer, 2020.

---

### Decision · Program_Chairs · 2022-01-20

**Decision:**

Accept (Poster)

**Comment:**

This paper proposes a few-shot learning method that uses Fisher information matrix-based task affinity. The experimental results show that the proposed method achieved better performance than existing methods. This paper is well-written. The newly proposed task affinity score is interesting. The experimental results and theoretical analysis support the effectiveness of the proposed method. The authors are encouraged to address the reviewers' concerns in the paper.  Although the distance between task representations is symmetric in neural processes, they do not use the symmetric distance for meta-learning. They input the task representations into the neural network, so the output can be asymmetric.